# Hyperuricemia Is an Early and Relatively Common Feature in Children with *HNF1B* Nephropathy but Its Utility as a Predictor of the Disease Is Limited

**DOI:** 10.3390/jcm10153265

**Published:** 2021-07-24

**Authors:** Marcin Kołbuc, Beata Bieniaś, Sandra Habbig, Mateusz F. Kołek, Maria Szczepańska, Katarzyna Kiliś-Pstrusińska, Anna Wasilewska, Piotr Adamczyk, Rafał Motyka, Marcin Tkaczyk, Przemysław Sikora, Bodo B. Beck, Marcin Zaniew

**Affiliations:** 1Department of Pediatrics, University of Zielona Góra, 65-417 Zielona Góra, Poland; rmotyka97@gmail.com; 2Department of Pediatric Nephrology, Medical University of Lublin, 20-059 Lublin, Poland; beatabienias@umlub.pl (B.B.); przemyslawsikora@umlub.pl (P.S.); 3Department of Pediatrics, University Hospital of Cologne, 50937 Cologne, Germany; sandra.habbig@uk-koeln.de; 4Department of Animal Physiology, Faculty of Biology, University of Warsaw, 00-927 Warsaw, Poland; m.kolek@biol.uw.edu.pl; 5Department of Pediatrics, Faculty of Medical Sciences in Zabrze, Medical University of Silesia in Katowice, 41-808 Zabrze, Poland; szczep57@poczta.onet.pl; 6Department of Paediatric Nephrology, Wrocław Medical University, 50-367 Wrocław, Poland; kilis@wp.pl; 7Department of Pediatric Nephrology, University Hospital, 15-089 Białystok, Poland; annwasil@interia.pl; 8Department of Pediatrics, Faculty of Medical Sciences in Katowice, Medical University of Silesia in Katowice, 40-752 Katowice, Poland; padamczyk@gczd.katowice.pl; 9Department of Pediatrics, Immunology and Nephrology, Polish Mother’s Memorial Hospital Research Institute, 93-338 Łódź, Poland; marcin.tkaczyk45@gmail.com; 10Institute of Human Genetics and Center for Molecular Medicine Cologne, University of Cologne, Faculty of Medicine and University Hospital Cologne, 50937 Cologne, Germany; bodo.beck@uk-koeln.de

**Keywords:** *HNF1B*, hyperuricemia, PTH, renal function, uric acid, FEUA

## Abstract

Background: Hyperuricemia is recognized as an important feature of nephropathy, associated with a mutation in the hepatocyte nuclear factor-1B *(HNF1B)* gene, and could serve as a useful marker of the disease. However, neither a causal relationship nor its predictive value have been proven. The purpose of this study was to assess this in children with renal malformations, both with (mut+) and without *HNF1B* mutations (mut-). Methods: We performed a retrospective analysis of clinical characteristics of pediatric patients tested for *HNF1B* mutations, collected in a national registry. Results: 108 children were included in the study, comprising 43 mut+ patients and 65 mut- subjects. Mean sUA was higher and hyperuricemia more prevalent (42.5% vs. 15.4%) in *HNF1B* carriers. The two groups were similar with respect to respect to age, sex, anthropometric parameters, hypertension, and renal function. Renal function, fractional excretion of uric acid and parathyroid hormone level were independent predictors of sUA. The potential of hyperuricemia to predict mutation was low, and addition of hyperuricemia to a multivariate logistic regression model did not increase its accuracy. Conclusions: Hyperuricemia is an early and common feature of *HNF1B* nephropathy. A strong association of sUA with renal function and parathyroid hormone limits its utility as a reliable marker to predict *HNF1B* mutation among patients with kidney anomalies.

## 1. Introduction

Hepatocyte nuclear factor 1β (*HNF1B*) is a critical transcription factor (encoded by the *HNF1B* gene) that regulates the development of the kidneys, pancreas, liver, and genital tract [1,2,3]. Its multi-organ expression results in a wide spectrum of renal and extra-renal manifestations in patients with *HNF1B*-associated disease, which includes renal cysts, multicystic dysplastic kidney, solitary kidney, agenesis of pancreas body and tail, maturity-onset diabetes in the young type 5, genital malformations, elevated liver enzymes, electrolyte abnormalities, primary hyperparathyroidism, gout, and epilepsy [4,5]. Hyperuricemia is a key hallmark of the disease that has been repeatedly reported in patients with *HNF1B* mutation [6,7,8,9,10,11,12,13,14,15]. Although this association is described in many studies, when compared to patients without *HNF1B* mutation, the frequency of hyperuricemia in patients with *HNF1B* mutation was not significantly different [10,11]. Thus, a causal relationship between *HNF1B* mutation and elevated serum uric acid (sUA) has not been proven to date.

Abnormalities in uric acid handling were first reported in 2003 in a kindred diagnosed with familial juvenile hyperuricemic nephropathy with an underlying mutation in the *HNF1B* gene [15]. Subsequently, analysis of sUA levels in patients with *HNF1B* mutations and healthy type 2 diabetic subjects without *HNF1B* mutation demonstrated significantly higher levels in *HNF1B* mutation carriers. However, the study found no significant difference between carriers and gender-matched controls with renal impairments of other causes. Despite this, the authors concluded that gout and hyperuricemia are general features of *HNF1B* mutations. Since this publication, the finding of elevated sUA levels in *HNF1B* mutation carriers has not been adequately analyzed in other cohorts [4]. It remains unclear whether raised sUA is a feature of *HNF1B*-related disease and is directly related to mutation, or if this symptom is a byproduct of chronic kidney disease (CKD), which could be present during the course of the disease.

A search of the literature revealed that the frequency of hyperuricemia varies significantly, depending mostly on the severity of CKD and age of the cohorts [6,7,8,9,10,11,12,13,14]. Importantly, in studies for which a combined cohort (i.e., children and adults) has been studied, the authors used higher upper limits for sUA, usually those applicable for adult patients. This could ultimately result in a low prevalence of hyperuricemia in children [10,14]. Others utilized reference data published in 1990, which are not reliable as these were generated from a small sample, and could be regarded as outdated [16]. Thus, we hypothesized that hyperuricemia might be misdiagnosed and under-recognized, especially in children, and this could limit investigations aimed at assessing its usefulness as a predictor of *HNF1B* disease. In fact, its diagnostic value has never been formally assessed.

We aimed to assess a causal relationship between elevated sUA level and *HNF1B* defects by comparing patients with and without mutations, and to investigate the predictive value of hyperuricemia for distinguishing *HNF1B* mutation carriers among patients with a range of renal anomalies.

## 2. Materials and Methods

### 2.1. Study Population

This is a multi-center, retrospective study of patients tested for *HNF1B* mutations, whose data were collected in the Polish Registry of Inherited Tubulopathies (POLtube) between July 2012 and July 2020. A suspicion of *HNF1B* nephropathy, based on a renal phenotype with additional extra-renal features, led to the initiation of *HNF1B* mutational analysis. The referral remained at clinicians’ discretion.

In total, 117 subjects suspected to have *HNF1B* nephropathy were recruited to the registry. Adults (*n* = 8) were excluded from the study to eliminate the effects of potential co-morbidities on the analyzed parameters, and to permit uniform analysis of hyperuricemia by applying the most recent pediatric norms [17]. One pediatric patient was excluded due to end-stage renal disease. Finally, the study group comprised 108 patients, of whom 43 patients were positive for *HNF1B* mutations (mut+), i.e., were carriers of a heterozygous point mutation or whole *HNF1B* gene deletion. The remaining patients (*n* = 65) were negative for *HNF1B* (mut-) and served as controls. The clinical and genetic characteristics of 14 of the mut+ patients have been reported previously [12,18,19].

### 2.2. Clinical Assessment and Definitions of Analyzed Parameters

Each subject’s anthropometrical and biochemical parameters and urinary indices, as well as details regarding family history of diabetes and/or renal anomalies, or renal and extra-renal phenotype and genotypes, were retrieved from the registry database and anonymized. Data were analyzed cross-sectionally at the time of molecular diagnosis for all patients, and at the time of the last visit in the mut+ group. In mut- patients, data were available only from the time of genetic testing. Renal phenotypes were characterized sonographically, similarly to Faguer et al. [20]. Extra-renal features had not been studied systematically in the cohort, but had been reported to the registry by the referring physician. Plasma and urine biochemistries were established using routine laboratory procedures, according to local policies. Both spot urine samples and 24 h urine collection were used to calculate urinary indices (fractional excretion of K^+^, Ca^2+^, Mg^2+^, and uric acid).

Serum Mg^2+^ (sMg) and sUA reference values for age and gender intervals proposed by Ridefelt et al. were applied with respect to hypomagnesemia and hyperuricemia definition [17]. Patients receiving supplementary oral Mg^2+^ (*n* = 12) and allopurinol (*n* = 8) were also considered as having low sMg or high sUA, respectively. Raw sMg/sUA values of these patients were not included in the analysis. Other parameters were defined as follows: hypokalemia as K^+^ level < 3.5 mmol/L, and hyperparathyroidism if the parathyroid hormone (PTH) level exceeded the local laboratory upper reference limit. Estimated glomerular filtration rate (eGFR) was calculated using the original or modified Schwartz equation when applicable [21,22]. Staging of CKD was based on eGFR and adopted from KDIGO (Kidney Disease: Improving Global Outcomes) guidelines [23], and was as follows: CKD stage I (eGFR > 90 mL/min/1.73 m^2^), stage II (eGFR between 60 and 89 mL/1.73 m^2^/min), stage III (eGFR between 30 and 59 mL/min/1.73 m^2^), CKD IV (eGFR between 15 and 29 mL/min/1.73 m^2^), and CKD V (eGFR < 15 mL/min/1.73 m^2^). Glucose metabolism disorders, i.e., impaired fasting glucose and diabetes mellitus, were defined according to the International Society of Pediatric and Adolescent Diabetes Guidelines [24]. Hypertension was defined as a systolic and/or diastolic blood pressure > 95th percentile appropriate for age, gender, and height. For children > 3 years of age, normative blood pressure values for Polish population were used [25,26], while children < 3 years reference values from the fourth report on blood pressure in children and adolescents [27]. Patients receiving antihypertensive drug was also defined as hypertensive. Standard deviation scores (SDS) for height and body mass index (BMI) were calculated from the World Health Organization growth charts (http://www.who.int/growthref/en, accessed on 26 August 2020). Growth impairment was defined as height-SDS < −2, and overweight as BMI-SDS > 1.

### 2.3. Molecular Analysis of HNF1B Gene

Genetic testing was performed as described elsewhere [12]. In brief, MLPA analysis was first performed to detect copy number variations (SALSA MLPA P241; MRC-Holland). If this result was negative, targeted Sanger sequencing of exons 1–9 of *HNF1B* was carried out. Written informed consent for the testing was obtained from parents and assent from children, where appropriate.

### 2.4. Statistical Analyses

Data were presented as means (95% confidence interval) for continuous variables and as counts (percentages) for categorical variables. Before proceeding to the analysis of differences, Shapiro–Wilk normality tests were performed on subgroups, and skewness was calculated. Equipotency of subgroups and variance homogeneity were determined if the normality assumption was met. To determine differences between patients with mutation at diagnosis and at follow-up, Student’s *t*-tests for dependent samples, Wilcoxon signed rank tests for numerical variables, and McNemar’s tests were performed. In order to compare this data with the control group, Student’s *t*-tests for independent samples, Mann–Whitney U tests and Chi-squared tests were used. Yates’ continuity correction and Welch’s correction for non-homogenous variances were applied if necessary. Separate regression analyses were used to determine the relationships between sUA and other continuous variables in the mut+ and mut- cohorts.

To predict concentrations of sUA, linear regression models were built. Firstly, univariate models were built to determine the most statistically significant predictors. Next, a multivariate model was built and simplified using a stepwise linear regression method. Only variables whose *p* value was less than 0.05 in the univariate analysis were included. The quality of the models was determined with the *R*^2^ coefficient. In order to predict mutation, an additional logistic regression model was implemented. Its quality was determined with the use of accuracy, sensitivity, and specificity coefficients. The variables included in the final model were determined using the information gain coefficient, based on conditional entropy. Data were split into training and test datasets. The level of significance was α = 0.05. Analysis was performed using R programming language, RStudio, and IBM SPSS Statistics 25.

## 3. Results

### 3.1. Characteristics of the Study Patients

The clinical characteristics of the cohorts are presented in Table 1. Among mut+ patients, *HNF1B* whole gene deletion was detected in 22/43 patients (51.2%). At the time of molecular testing, both cohorts were comparable with respect to age, sex distribution, anthropometric parameters, prevalence of hypertension, and eGFR. In both groups, renal function was well preserved; The majority of patients had CKD stage 1 or stage 2 (91.2% and 94.1% of mut+ and mut- cohorts, respectively). As expected, well-known features of *HNF1B*-related disease, i.e., hypomagnesemia (65%) and hyperparathyroidism (38.5%) were more frequently found than in mut- subjects. The two cohorts were also different with respect to impairments in glucose metabolism (30% vs. 7.7% for mut+ and mut-, respectively), pancreatic anomalies (20.9%; exclusively found in mut+ patients), and liver involvement (14.3% vs. 1.5%).

All patients presented with renal anomalies on ultrasound. A phenotypic comparison of the two cohorts is presented in Table 2. Cystic kidney disease was the most common renal phenotype in both groups. In those with *HNF1B* mutations, increased renal echogenicity and multicystic dysplastic kidney disease were more prevalent.

For a great number of mut+ patients, follow-up data were also available. The mean record duration was 40.2 months (95% CI; 30.9–49.6). On follow-up, no significant changes were observed in these patients compared with the baseline data.

### 3.2. Serum Uric Acid Concentration

At the time of molecular testing, the mean sUA was higher in mut+ than in mut- subjects (5.74 mg/dL (95% CI; 5.22–6.27) vs. 4.87 mg/dL (95% CI; 4.47–5.28), *p* = 0.006). Hyperuricemia was more frequent in mut+ patients than in mut- cases (42.5% vs. 15.4%, *p* = 0.002). During the follow-up period, another four patients developed hyperuricemia. All sUA measurements for boys and girls are plotted in Figure 1. No clinical symptoms of gout were present in either cohort. Among mut+ patients, an elevated level of sUA was more frequent in those harboring point mutations than deletions of the *HNF1B* gene (57.1% vs. 29.7%, *p* < 0.019). No influence of any renal feature/phenotype on hyperuricemia prevalence was found.

A sensitivity analysis was performed after exclusion of patients with impairments in glucose metabolism. Raised sUA was present in 35.7% (in 10/28 patients) vs. 13.3% (in 8/60 patients; *p* = 0.015) of the mut+ and mut- groups, respectively. There was also a tendency for a higher mean sUA (5.33 mg/dL (95% CI; 4.89–5.76) vs. 4.75 mg/dL (95% CI; 4.34–5.15), *p* = 0.052), between mut+ and mut- subjects, respectively.

Due to the age dependence of sUA, the mut+ and mut- cohorts were arbitrarily divided into two age groups (0–9 and 10–18 years) and compared, with respect to sUA, eGFR, BMI-SDS, fractional excretion of uric acid (FEUA), and the frequency of hyperuricemia. Higher mean values of sUA were observed in patients harboring *HNF1B* mutations compared with mut- subjects within both age intervals (*p* = 0.003 and 0.052 for 0–9 and 10–18 years, respectively). Elevated sUA was already present at an early age, i.e., in 37.5% of *HNF1B* patients in the first decade versus 50% in the second decade of life (Table 3). In this regard, the difference between mut+ and mut- cohorts was more pronounced in younger children. Notably, there were no differences between the remaining analyzed parameters that could explain the difference in sUA. Figure 2 shows the relationships between sUA and eGFR (panel a), and between sUA and FEUA (panel b) for the mut+ and mut- groups separately.

### 3.3. Fractional Excretion of Uric Acid

FEUA was comparable between the mut+ and mut- groups (6.88% (95% CI; 5.27–8.5) vs. 6.16% (95% CI; 5.59–6.74), *p* = 0.888), respectively) when analyzing the entire cohort, as well as within age sub-groups (Table 3). Among mut+ patients, there was no difference in FEUA depending on the mutation type. Figure 3 shows the relationships between FEUA and age (panel a), and between FEUA and eGFR (panel b).

### 3.4. Determinants of Serum Uric Acid

In the univariate linear regression models, age, eGFR, CKD, sMg, hypomagnesemia, fractional excretion of Mg^2+^, PTH, hyperparathyroidism, glucose metabolism disorders, hypertension, *HNF1B* mutation, and FEUA were all significantly associated with sUA (Appendix A). However, in a multivariate linear stepwise regression model, only eGFR, FEUA, and PTH were independent predictive variables of sUA (*R*^2^ = 0.85, F = 41.47, *p* < 0.001) (Table 4). sUA correlated positively with PTH (*R*^2^ = 0.18, *p* < 0.001).

### 3.5. Hyperuricemia as a Predictor of HNF1B Mutation

The potential of hyperuricemia for mutation prediction was tested in a model with all parameters that were significantly greater in the mut+ group compared with the mut- group of patients. A model including hypomagnesemia and hyperparathyroidism showed an accuracy of 85% (sensitivity: 83%, specificity: 86%). Adding hyperuricemia to the model did not increase the accuracy (79%; sensitivity: 77%, specificity: 82%). Information gain, which represents the selective potential of each parameter, was the lowest for hyperuricemia (0.34 compared with 0.99 and 0.63 for hyperparathyroidism and hypomagnesemia, respectively).

## 4. Discussion

This is the first study to comprehensively assess uric acid handling in a large cohort of pediatric patients with renal anomalies. We show that hyperuricemia is a relatively common feature in patients with *HNF1B*-related kidney disease, already present in early childhood. The age of manifestation of hyperuricemia is in line with previous observations [7,12]. Importantly, the frequency of hyperuricemia was compared to that present in counterparts who are negative for *HNF1B* mutations. Here, we showed a significant difference, which cannot be due to the effects of age, renal function and phenotype, body mass, gender, or hypertension, as the two cohorts we analyzed were comparable with respect to these parameters. Furthermore, a sensitivity analysis, which excluded individuals with glucose metabolism disorders, retained the difference. This result is in concordance with the findings of Bingham et al. [15], and this suggests that hyperuricemia is not just secondary to hyperglycemia/diabetes. Based on our findings, we can conclude that elevated sUA is causally related to *HNF1B* defect.

Although plenty of data exist on the presence of this abnormality in *HNF1B* patients, there has been a lack of good evidence on this association to date. For example, to exclude the effect of renal dysfunction, in a study by Bingham et al. [15], a comparison of sUA levels was made between *HNF1B* subjects (*n* = 8) with gender-matched subjects with renal impairment of other causes (*n* = 32). Notably, the analysis did not show significance. Although the authors claimed hyperuricemia is a feature of *HNF1B* mutation, the conclusions are vague due to the small sample size. In other studies, the frequency of hyperuricemia was not significantly different compared to *HNF1B* negative patients [10,11]. The lack of differences could possibly be explained by some differences in patient age and severity of CKD. For these reasons, our analysis was restricted only to children, who are free of co-morbidities and characterized by no alcohol consumption or multidrug use. Unfortunately, in our study we cannot exclude other variables that may impact sUA, with these three parameters explaining 85% of the total model variance. Lack of information about diet, physical activity, and blood lipids is a limitation of our study.

It is well known that elevated sUA is attributed to renal dysfunction. sUA is cleared by the kidney, and its level rises with declining kidney function [28]. Our data reflect this in that eGFR was a strong independent predictor of sUA along with FEUA and PTH. Unlike renal dysfunction, an independent relationship between sUA and PTH was quite unexpected. However, a literature search showed that there are several studies, which indicate hyperuricemia is more common among patients with primary hyperparathyroidism [29,30]. An evidence supporting the effect of PTH on sUA was delivered by Hui et al. in a large national population of adults from the USA [31]. By using data from 8316 participants, authors proved that serum PTH levels are independently associated with sUA levels and the frequency of hyperuricemia at the population level. This association is independent of renal function. Elevated PTH is thought to reduce renal uric acid excretion, but the exact mechanism is not known [32]. Our finding provides further evidence on this relationship. Given our observations, one has to take into consideration not only GFR, but also PTH when assessing hyperuricemia. This, however, might be a limitation when considering hyperuricemia as a reliable predictor of *HNF1B* nephropathy.

As we have shown, appropriate norms are instrumental in the assessment of hypomagnesemia in children with *HNF1B* mutations [18]. In this respect, applying the most recent reference limits for sUA could be regarded as a strength of the study. We used age- and gender-specific upper limits for sUA, as norms change when the body grows. A comprehensive review article by Kubota [33] shows that there are many conditions leading to hyperuricemia in children and adolescents. Among these, a range of both congenital and acute diseases, as well as lifestyle-related disorders, are listed as strongly associated with elevated sUA. Hyperuricemia may be encountered across the entire age spectrum of pediatric patients, and appropriate reference values should therefore be applied. Ridefelt et al. [17] defined these by calculating 2.5th and 97.5th percentiles of sUA in a group of 1998 healthy children and adolescents from Sweden and Denmark. Three age subgroups were distinguished, however differences between boys and girls were only observed in children > 10 years of age.

Importantly, in our study hyperuricemia was present in 42.5% of *HNF1B* mutation carriers. Notably, the cohort was characterized by a mild stage of CKD, which excludes a significant effect of renal insufficiency from impacting the results. In this context, the hyperuricemia prevalence is within the expected range; a literature review shows that, to date, prevalence has been recorded in the range of 11–71%. This variation could be mostly due to differences in age or renal function (Table 5). Unfortunately, some authors do not provide reference values and/or the data are not presented separately for children and adults (i.e., one cut-off value is applied for the mixed cohort) or the number of patients with available data on hyperuricemia is too small to draw conclusions from. The above limitations might have resulted in under-recognition of this parameter for predicting *HNF1B* mutation. Indeed, the screening tool to select patients eligible for genetic analysis of the *HNF1B* gene (HNF1B score) does not recognize hyperuricemia as a weighted parameter [20]. Instead, only gout is considered an important symptom [20], which in fact does not usually present until adulthood. In this respect, we provide the evidence that hyperuricemia is not an accurate predictor and, consequently, its application in HNF1B score might be of little value. In comparison to hypomagnesemia and hyperparathyroidism, both of which were significantly higher in the mut+ versus mut- group, the information gain, which describes the power of a variable, was rather low and did not significantly improve the model fit. One could expect that additional assessment of FEUA could improve the prediction. However, we demonstrated no difference in FEUA between the two groups (Figure 3a), which is concordant with the findings from the study by Bingham et al. [15]. Furthermore, FEUA values in patients with mutations in the uromodulin gene overlap slightly with those obtained from controls with normal kidney function [34], which confirms it is unreliable measure, and eliminates FEUA as a discriminative marker.

Among other laboratory observations that could be used as predictors of *HNF1B*, hypomagnesemia seems to be particularly valuable, as confirmed by our study. Notably, when we utilized age- and gender-dependent norms for sMg [17], hypomagnesemia was present in 65% of mut+ patients, which sharply contrasted with 4.8% in the mut- group. Here, we demonstrated for the first time that applying the appropriate reference lower limits for sMg instead of one cut-off value (sMg < 0.7 mmol/L) for the entire cohort we achieved a good discrimination between mut+ and mut- cohorts. Despite this, it is highly reasonable to use a combination of easily available laboratory markers instead of one in creation of a *HNF1B* score, which could be used as a tool for *HNF1B* mutation prediction in children. Interestingly, we showed that hyperparathyroidism, already described by others as a feature of *HNF1B* nephropathy [35], could also be instrumental, but probably only in those with good renal function. There is a concern that in a more severe CKD population, secondary hyperparathyroidism will distort this relationship. Importantly, the HNF1B score by Faguer et al. [20] was developed for both adults and children, which makes this score imperfect, especially in young children, wrongly predicting the absence of a mutation [14]. In this respect, there is a need for revision of the existing score or creation of a new HNF1B score or a diagnostic algorithm that could be applicable specifically in children. Based on our findings, hyperuricemia should not be highly valued.

We are aware of some limitations relating to the retrospective nature of this study. For some patients, the data were not complete, and parameters were obtained at unequal time points. As the patients were recruited from different centers, the method of sUA assessment was non-uniform, and extra-renal abnormalities were not studied formally. On the other hand, there are some strengths that need to be stressed. These are: a relatively large patient cohort free of co-morbidities and with only few patients requiring pharmacological treatment, and a well-matched control group of patients with urinary tract malformations and negative for *HNF1B* mutations. As mentioned before, our study was performed in a cohort with mildly affected renal function, which eliminates to some extent a confounding effect of renal dysfunction. Finally, the application of the most recent reference data for sUA is of great value. On the other hand, due to lack of local sUA reference values, we applied those derived from a Nordic population. This could impact our results on the frequency of hyperuricemia, as sUA levels may differ between populations and ethnicities. Thus, our findings should be validated in an international cohort.

## 5. Conclusions

We demonstrated that hyperuricemia is an early and prevalent feature in children with *HNF1B* nephropathy when compared to well-matched mutation negative patients. Thus, we provide evidence that this abnormality is causally related to *HNF1B* defect. A strong association of sUA with renal function and PTH limits a clinical assessment of hyperuricemia. Although our results derive from pediatric population, it especially applies to adults who are at a greater risk of CKD. As we showed, the utility of hyperuricemia as a predictor of *HNF1B*-related disease is limited, and should not be relied upon when selecting patients for genetic testing. We also propose that assessment of FEUA would not be helpful. The presence of hyperparathyroidism seems to be discriminative, however further studies are needed to prove this observation.

## Figures and Tables

**Figure 1 jcm-10-03265-f001:**
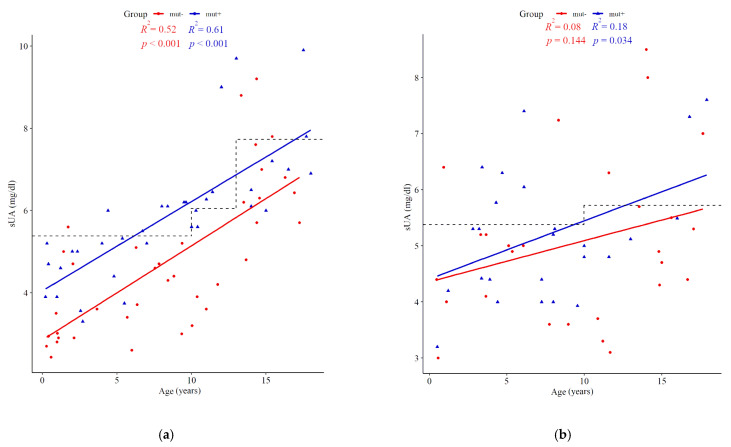
Scatter plots of serum uric acid (sUA) concentrations showing trends with age in children harboring *HNF1B* mutations (mut+, blue triangles) versus those without mutations (red dots). Boys (**a**) and girls (**b**) are presented separately with age-adjusted upper limits of reference values applied as dashed lines.

**Figure 2 jcm-10-03265-f002:**
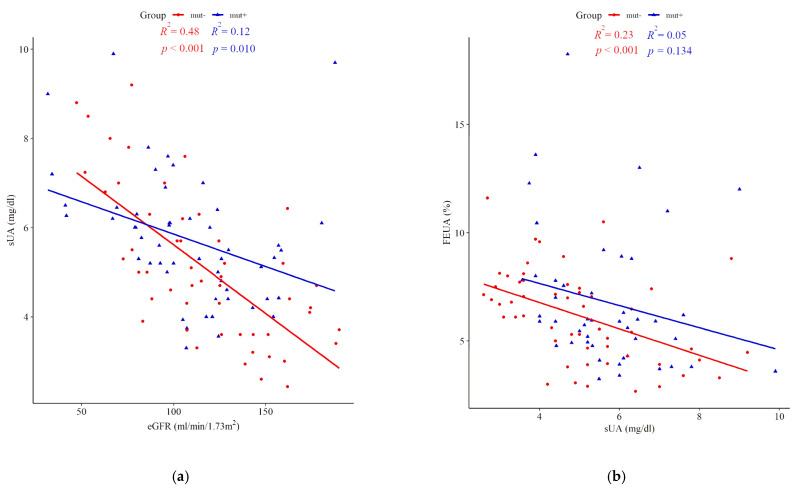
Correlations between serum uric acid (sUA) concentrations and estimated glomerular filtration rate (eGFR) (**a**) and fractional excretion of uric acid (FEUA) (**b**) in individuals with *HNF1B* mutations (mut+, blue triangles) and those who were negative for mutations (mut-, red dots).

**Figure 3 jcm-10-03265-f003:**
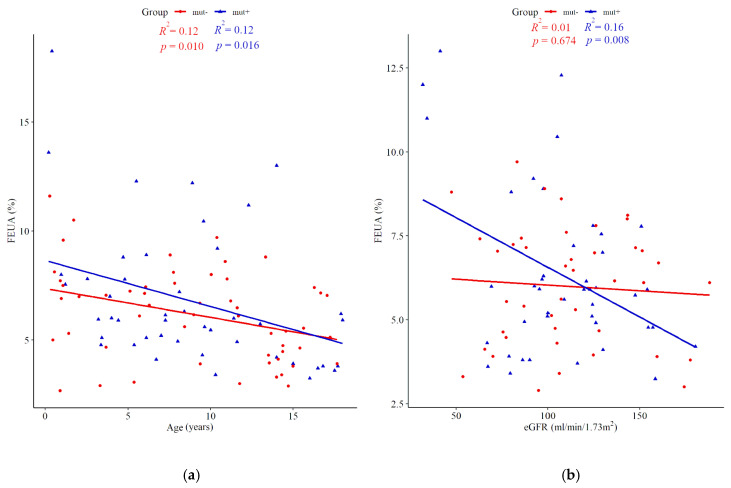
Fractional excretion of uric acid (FEUA) according to age (**a**) versus estimated glomerular filtration rate (eGFR) (**b**) in the *HNF1B* positive cohort (mut+, blue triangles) and *HNF1B* negative cohort (mut-, red dots).

**Table 1 jcm-10-03265-t001:** Characteristics of the study cohorts.

Variable	mut+ at Diagnosis	mut-	*p* Value
Age (years)	7.84 (6.10–9.58)	8.74 (7.34–10.14)	0.436
Gender (M/F)	24/16 (60.5–39.5)	38/27 (58.5–41.5)	0.876
Height-SDS	0.04 (−0.53–0.61)	0.10 (−0.24–0.45)	0.857
Short stature	4 (11.1)	2 (3.8)	0.355 ^a^
BMI-SDS	0.49 (−0.03–1.00)	0.34 (0.03–0.66)	0.324
Overweight	13 (36.1)	13 (24.5)	0.238
eGFR (mL/min/1.73 m^2^)	104.09 (90.40–117.78)	115 (104.32–126.03)	0.210
CKD			---
1	21 (63.6)	36 (70.6)	0.505
2	9 (27.3)	12 (23.5)	0.699
3	3 (9.1)	3 (5.9)	0.577
4	---	---	---
5	---	---	---
sMg (mmol/L)	0.72 (0.69–0.76)	0.86 (0.84–0.88)	<0.001
Hypomagnesemia	26 (65.0)	3 (4.8)	<0.001
sUA (mg/dL)	5.74 (5.22–6.27)	4.87 (4.47–5.28)	0.006
Hyperuricemia	17 (42.5)	10 (15.40)	0.002
sK (mmol/L)	4.43 (4.26–4.60)	4.64 (4.51–4.78)	0.041
Hypokalemia	0 (0.0)	0 (0.0)	---
PTH (pg/mL)	67.78 (45.50–90.06)	33.46 (27.09–39.82)	0.001
Hyperparathyroidism	10 (38.5)	1 (3.0)	0.001
FEMg (%)	6.89 (5.54–8.24)	3.97 (3.35–4.60)	<0.001
FEUA (%)	6.88 (5.27–8.50)	6.16 (5.59–6.74)	0.888
FECa (%)	0.65 (0.29–1.01)	0.51 (0.35–0.66)	0.754
FEK (%)	7.48 (4.61–10.36)	5.68 (4.72–6.65)	0.149
Positive family history ^b^	21 (51.2)	27 (58.7)	0.484
IFG/DM	10/2 (30.0)	4/1 (7.70)	0.003
Pancreatic anomalies	9 (20.9)	0 (0)	<0.001 ^a^
Liver involvement	6 (14.3)	1 (1.5)	0.028 ^a^
Genital tract anomaly	2 (4.7)	4 (6.2)	0.305 ^a^
Developmental/speech delay	3 (7.0)	1 (1.5)	0.143
Hypertension	11 (27.5) ^c^	10 (15.4) ^d^	0.529

Data are presented as numbers (percentages) or as means (95% confidence interval). BMI-SDS, body mass index standard deviation score; CKD, chronic kidney disease; DM, diabetes mellitus; eGFR, estimated glomerular filtration rate; F, female; FECa, fractional excretion of Ca^2+^; FEK, fractional excretion of K^+^; FEMg, fractional excretion of Mg^2+^; FEUA, fractional excretion of uric acid; Height-SDS, height standard deviation score; IFG, impaired fasting glucose; M, male; mut+, *HNF1B* positive patients; mut-, *HNF1B* negative patients; PTH, parathyroid hormone; sK, serum K^+^; sMg, serum Mg^2+^; and sUA, serum uric acid. ^a^ Yates continuity correction to Chi-squared test was performed; ^b^ family member with either diabetes mellitus and/or structural kidney disease; ^c^ medications (*n*): angiotensin-converting enzyme inhibitors (7), Ca^2+^ channel blockers (5), β-blocker (1); ^d^ medications (*n*): angiotensin-converting enzyme inhibitors (8); angiotensin receptor blocker (1), and Ca^2+^ channel blocker (1).

**Table 2 jcm-10-03265-t002:** Renal phenotype with respect to *HNF1B* mutation status.

	HNF1B+	HNF1B-	*p* Value
Renal hyperechogenicity	27 (73.0)	10 (27.0)	<0.001
Renal cysts	26 (43.4)	34 (56.7)	0.359
Multicystic dysplastic kidney	17 (60.7)	11 (39.3)	0.008
Renal hypoplasia/hypodysplasia	8 (57.1)	6 (42.9)	0.147
Urinary tract malformations	11 (91.7) ^a^	1 (8.3) ^b^	<0.001
Solitary kidney	5 (27.8)	13 (72.2)	0.267

Data were presented as numbers of patients (percentages). ^a^ pelviectasis (*n* = 7), posterior urethral valve-like partial bladder outlet obstruction (*n* = 1), kidney ectopy (*n* = 1), kidney malrotation (*n* = 1), and vesicoureteral reflux (*n* = 1). ^b^ ureteral dilatation.

**Table 3 jcm-10-03265-t003:** Differences in serum uric acid levels, frequency of hyperuricemia and fractional excretion of uric acid between *HNF1B* positive and negative patients presented for age-specific intervals.

Parameter	Age Group	HNF1B+	HNF1B-	*p* Value
BMI-SDS	0–9	0.31 (−0.19–0.81; *n* = 37)	0.32 (−0.26–0.90; *n* = 21)	0.992
	10–18	0.41 (−0.03–0.86; *n* = 31)	0.36 (−0.03–0.74; *n* = 32)	0.846
eGFR (mL/min/1.73 m^2^)	1–9	112.28 (103.80–120.76; *n* = 33)	128.89 (109.05–148.72; *n* = 17)	0.119
	10–18	95.01 (80.59–109.43; *n* = 32)	107.89 (94.96–120.81; *n* = 32)	0.180
sUA (mg/dL)	0–9	4.93 (4.59–5.26; *n* = 35)	4.14 (3.73–4.54; *n* = 33)	0.003
	10–18	6.50 (5.89–7.11; *n* = 25)	5.63 (5.00–6.26; *n* = 32)	0.052
Hyperuricemia (%)	0–9	37.5% (*n* = 24)	8.6% (*n* = 35)	0.017
	10–18	50% (*n* = 16)	23.3% (*n* = 30)	0.066
FEUA (%)	0–9	7.33 (6.01–8.65; *n* = 25)	6.82 (5.92–7.72; *n* = 25)	0.511
	10–18	6.16 (4.77–7.55; *n* = 21)	5.60 (4.87–6.32; *n* = 29)	0.425

Data are presented as means (95% confidence interval) or percentages. BMI-SDS, body mass index standard deviation score; eGFR, estimated glomerular filtration rate; FEUA, fractional excretion of uric acid; and sUA, serum uric acid.

**Table 4 jcm-10-03265-t004:** Independent predictors of serum uric acid.

Parameter	B	SE	Beta	t	F	*R* ^2^
(Constant)	9.95	0.85	---	11.72 ***	41.47 ***	0.85
eGFR	−0.03	0.01	−0.60	−6.31 ***		
PTH	0.02	0.00	0.44	4.75 ***		
FEUA	−0.31	0.07	−0.36	−4.24 ***		

*** *p* < 0.001; eGFR, estimated glomerular filtration rate; FEUA, fractional excretion of uric acid; and PTH, parathyroid hormone.

**Table 5 jcm-10-03265-t005:** Literature overview of the frequency of hyperuricemia in different patient cohorts.

Study Group	Hyperuricemia				Renal Function in Hyperuricemic Patients (Number of Cases)	Reference Values for Hyperuricemia
	HNF1B+		HNF1B-			
	Children	Adults	Children	Adults		
Ulinski et al. (2006) [6]	7/23 ^a^ (30%)	-	-		not specified	not given
Decramer et al. (2007) [7]	11/18 (61%)	-	-		CKD I (3), CKD II (2), CKD III (5), CKD 5 (1)	not given
Adalat et al. (2009) [8]	10/14 (71%)	-	7/15 (47%)	-	CKD ≤ III only for both *HNF1B*+ and *HNF1B-*	age-dependent reference limits
Heidet et al. (2010) [9]	12/75 (16%) ^b^		-		CKD I (3), CKD II (1), CKD III (1), CKD V (1), no data on remaining 6 cases	not given
Raaijmakers et.al (2014) [10]	4/20 (20%)		41/185 (22.2%)		CKD I–III (in all *HNF1B*+ cohort), no data on renal function in *HNF1B-*	>5.7 mg/dL in females, >7 mg/dL in males, irrespective of age
Madariaga et al. (2018) [11]	6/17 (27%)	0/4	12/36 (33%)	-	CKD I (1), CKD II (1), CKD III (2), CKD V (2) in *HNF1B*+, no data on renal function in *HNF1B*-	not given
Okorn et al. (2019) [12]	19/52 (37%)	-	-		CKD I (10), CKD II (5),CKD IV (2), CKD V (2)	age-dependent reference limits
Nagano et al. (2019) [13]	2/18 (11%)	4/13 (31%)	-		CKD III (5), CKD IV (1)	sUA > 7 mg/dL, irrespective of age and sex
Lim et al. (2020) [14]	8/11 (73%)	3/3 (100%)	-		CKD I (2), CKD II (4), CKD III (4), CKD V (1)	sUA > 7 mg/dL, irrespective of age and sex
Our study (2021)	17/40 (42.5%)	-	10/65 (15.4%)	-	CKD I (7), CKD II (8), CKD III (2)	age- and sex-appropriate norms

CKD, chronic kidney disease; HNF1B+, *HNF1B* positive patients; HNF1B-, *HNF1B* negative patients; and sUA, serum uric acid. ^a^ moderately elevated level of serum uric acid (<1.5 times the upper normal level). ^b^ gout and/or hyperuricemia.

## Data Availability

The data presented in this study are available on request from the corresponding author.

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
