# Peer review of "Hyperuricemia Is an Early and Relatively Common Feature in Children with HNF1B Nephropathy but Its Utility as a Predictor of the Disease Is Limited"

_jcm, 2021, doi:10.3390/jcm10153265_

Round 1
Reviewer 1 Report
The authors address the relevant question of whether kidney disease associated with the HNF1B mutation is causally related to hyperuricemia. Since this study deals with a rare disease, the study has a retrospective design and enrolled 108 children, 43 of them had the HNF1B mutation and these had follow-up data available. In the other patients, although clinically suspected, the disease was not confirmed. For a rare disease such as this, this is most likely the best possible study design.
The study provides the characteristics of the HNF1B mutation positve patients and how they discriminate from the control group. This is very important reference data. While hyperuricemia was associated with the mutation, it turned out not to be a helpful diagnostic marker.
This is important and clarifying work. The cohort is derived from the Polish Registry of Inherited Tubulopathies. It is most likely quite homogeneous in terms of ethnicity. The authors might consider adding a sentence regarding its generalizability to other populations.
Author Response
We fully agree with the Reviewer. Indeed, sUA levels may differ between populations and ethnicities, thus our results should not be generalized. As such a validation study in a large international cohort is required. This limitation was added to the Discussion section.
Reviewer 2 Report
- How was eGFR calculated in this case? these are pediatric patients.
- How was CKD staging diagnosed in this study? In addition, there is no data available on urinalysis microscopy findings e.g. the amount of protein and albumin to creatinine ratio; these are important data, as well as other urinalysis findings
- How was hypertension diagnosed needed to be clarified given these are pediatric populations
- Urinary tract malformations need to be additionally described; how severe they are? provide additional details are needed.
- Figures; red and blue colors should be darker; light red and light blue should be discouraged because they are difficult to read
Author Response
1. How was eGFR calculated in this case? these are pediatric patients.
Response:
This is already mentioned in the Methods section that estimated glomerular filtration rate was calculated using the original or modified Schwartz equation depending on applied creatinine measurement method.
2. How was CKD staging diagnosed in this study? In addition, there is no data available on urinalysis microscopy findings e.g. the amount of protein and albumin to creatinine ratio; these are important data, as well as other urinalysis findings
Response:
Staging of CKD was based on eGFR and adopted from KDIGO (Kidney Disease: Improving Global Outcomes) guidelines, and was as follows: CKD stage I (eGFR > 90 ml/min/1.73m2), stage II (eGFR between 60 and 89 ml/1.73 m2/min), stage III (eGFR between 30 and 59 ml/min/1.73 m2), CKD IV (eGFR between 15 and 29 ml/min/1.73 m2), and CKD V (eGFR < 15 ml/min/1.73 m2). This information has been added.
We fully agree that data on urinary sediment and proteinuria and/or albuminuria are very important. Unfortunately, these data were not available for analysis. Due to a retrospective nature of this study, when referring a patient to the Registry, these data were not provided by clinicians as not being crucial for diagnosis of HNF1B.
3. How was hypertension diagnosed needed to be clarified given these are pediatric populations
Response:
Hypertension was defined as a systolic and/or diastolic blood pressure > 95th percentile appropriate for age, gender and height for Polish population [KuÅ‚aga Z, Litwin M, Grajda A, KuÅ‚aga K, Gurzkowska B, Góźdź M, Pan H; OLAF Study Group. Oscillometric blood pressure percentiles for Polish normal-weight school-aged children and adolescents. J Hypertens. 2012 Oct;30(10):1942-54; Grajda A, KuÅ‚aga Z, Gurzkowska B, WojtyÅ‚o M, Góźdź M, Litwin M. Preschool children blood pressure percentiles by age and height. J Hum Hypertens. 2017 Jun;31(6):400-408]. In children below the age of 3 years, normative values for blood pressure from the fourth report on blood pressure in children and adolescents were used [National High Blood Pressure Education Program Working Group on High Blood Pressure in Children and Adolescents. The fourth report on the diagnosis, evaluation, and treatment of high blood pressure in children and adolescents. Pediatrics. 2004 Aug;114(2 Suppl 4th Report):555-76].
Patient receiving antihypertensive drug was also found to be hypertensive.
We have clarified this issue in the manuscript, and have added new references.
4. Urinary tract malformations need to be additionally described; how severe they are? provide additional details are needed.
Response:
In all HNF1B positive and negative patients renal anomalies predominated. Urinary tract malformations were defined as additional findings noticed on US scan or reported by clinicians at the time of referral. Unfortunately, we are not able to provide any information on the severity of the malformations.
We have provided more detailed information on this in Table 3 footnote.
5.Figures; red and blue colors should be darker; light red and light blue should be discouraged because they are difficult to read
Response:
Suggested changes were made and old figures replaced in the manuscript.
Round 2
Reviewer 2 Report
the authors have addressed the raised issues, no further comment
Author Response
We really appreciate all concerns raised and we are happy that our replies are satisfactory.